# Chronic Viral Hepatitis Signifies the Association of Premixed Insulin Analogues with Liver Cancer Risks: A Nationwide Population-Based Study

**DOI:** 10.3390/ijerph16122097

**Published:** 2019-06-13

**Authors:** Chien-Hsieh Chiang, Chia-Sheng Kuo, Wan-Wan Lin, Jun-Han Su, Jin-De Chen, Kuo-Chin Huang

**Affiliations:** 1Department of Family Medicine, National Taiwan University Hospital & College of Medicine, Taipei 100, Taiwan; jiansie@ntu.edu.tw; 2Graduate Institute of Pharmacology, National Taiwan University College of Medicine, Taipei 100, Taiwan; wwllaura1119@ntu.edu.tw; 3Department of Community and Family Medicine, National Taiwan University Hospital Yunlin Branch, Yunlin 640, Taiwan; 4Department of Community and Family Medicine, National Taiwan University Hospital Bei-Hu Branch, Taipei 108, Taiwan; omigodokuo@gmail.com; 5Department of Molecular and Cellular Biology, Harvard University, Cambridge, MA 02138, USA; junhansu@g.harvard.edu; 6Division of Gastroenterology and Hepatology, Department of Internal Medicine, National Taiwan University Hospital Bei-Hu Branch, Taipei 108, Taiwan; d900321@yahoo.com.tw; 7Hepatitis Research Center, National Taiwan University Hospital, Taipei 100, Taiwan

**Keywords:** hepatocellular carcinoma, chronic hepatitis B, chronic hepatitis C, population-based study, type 2 diabetes mellitus

## Abstract

This study sought to determine whether chronic hepatitis B or C would modify the association between insulin analogues and hepatocellular carcinoma (HCC) risks. We conducted a nationwide nested case-control study for HCC cases and matched controls from 2003 to 2013 among newly diagnosed type 2 diabetes patients on any antidiabetic agents in Taiwan before and after exclusion of chronic viral hepatitis, respectively. A total of 5832 and 1237 HCC cases were identified before and after exclusion of chronic viral hepatitis, respectively. Incident HCC risks were positively associated with any use of premixed insulin analogues (adjusted odds ratio (OR), 1.27; 95% CI 1.04 to 1.55) among total participants, especially among current users (adjusted OR, 1.45; 95% CI 1.12 to 1.89). However, the association between HCC occurrence and premixed insulin analogues diminished among participants without chronic viral hepatitis (adjusted OR, 1.35; 95% CI 0.92 to 1.98). We also observed a significant multiplicative interaction between chronic viral hepatitis and premixed insulin analogues on HCC risks (*P* = 0.010). Conclusions: Chronic viral hepatitis signifies the role of premixed insulin analogues in HCC oncogenesis. We recommend a closer liver surveillance among patients prescribed premixed insulin analogues with concomitant chronic viral hepatitis.

## 1. Introduction

Hepatocellular carcinoma (HCC) ranks as a leading cause of cancer death, especially in Asia-Pacific regions [1]. Risk factors for HCC include not only chronic viral hepatitis and alcohol consumption [2,3,4,5], but also diabetes mellitus [6,7,8,9,10,11,12,13,14]. Existing systematic reviews have shown that diabetes mellitus was associated with increased risks of HCC and decreased disease-free survival [15,16]. A meta-analysis enrolling 21 cohort studies also demonstrated that type 2 diabetes increased the risk of HCC among patients with chronic liver disease, cirrhosis, chronic hepatitis C virus infection, and chronic hepatitis B virus infection [17]. Furthermore, investigators should clarify the independent role of antidiabetic agents in HCC development. There was research analyzing the database of National Health Insurance in Korea for associations of antidiabetic drugs with HCC [18]. The use of commonly prescribed insulin has been reported to increase the risk of HCC [19,20]. All insulin analogues are covered by health insurance in Taiwan and commonly used for glycemic control in general practice [21,22]. The safety concerns should neither be disregarded nor misunderstood. Nonetheless, most studies did not evaluate whether chronic viral hepatitis modifies the effects of insulin analogue on HCC. Therefore, we aimed to test the hypothesis that chronic viral hepatitis might modify the reported associations between exogenous insulin analogues and HCC risks in a nationwide population with newly diagnosed diabetes on any antidiabetic agents, before and after excluding chronic viral hepatitis.

## 2. Materials and Methods

### 2.1. Data Source

The National Health Insurance database in Taiwan includes complete outpatient visits, hospital stays, prescriptions, and disease and vital status for 99% of the country’s population (approximately 23 million). The current analyses linked several large computerized claims datasets with the National Death Registry and the National Cancer Registry through the use of birth dates and civil identification numbers unique to each beneficiary. The datasets are protected under government laws and regulations, allowing no public access. The protocol had been reviewed by the Research Ethics Committee A of the National Taiwan University Hospital and complied with the categories of exemption from informed consent.

### 2.2. Study Design and Population

After critically designing the protocol based on our hypothesis and obtaining approval from the Minister of Health and Welfare, we retrospectively generated the study participants by computer linkage. We accessed the nationwide population for eligible participants, defined as newly diagnosed type 2 diabetes patients aged 30 to 90 years on any antidiabetic agents from the year of drug availability (2003 for premixed insulin and 2004 for insulin glargine or detemir) to 2013. We excluded participants if they did not have continuous insurance coverage 12 months before the index date; had a pre-existing diagnosis of diabetes or prescription of antidiabetic drugs within 12 months of study entry; or had a pre-existing diagnosis of type 1 diabetes or any cancer before the index date. Patients with a diagnosis of chronic hepatitis B or C infection or any relevant treatments were excluded from the corresponding analyses. Figure 1 illustrates flowcharts of the study participants before and after, excluding chronic viral hepatitis from 2003 to 2013. The flowcharts of the study participants from 2004 to 2013 are shown in Appendix A.

### 2.3. Cases Ascertainment and Controls Selection

Participants who had a first diagnosis of HCC with the ICD-9-CM code 155.0 were included as cases, confirmed by linking to Taiwan’s National Cancer Registry and National Death Registry. For each case in the pre-specified cohorts described above, a risk-set sampling matched on age (within 5 years), gender, and follow-up duration was used to find four controls in the same cohort.

### 2.4. Exposure Ascertainment and Covariate Adjustment

The exposure of interest was use of insulin analogues before HCC occurrence. We collected information on drug types, dates of prescription, supply days, and total dosage dispensed from the outpatient pharmacy prescription database. Participants could receive a combination of these study insulin analogues or switch from one class to another. Exposure to these medications was treated as a time-dependent variable in the analysis. To investigate the influence of dose, we had calculated the cumulative use of each class of basal and premixed insulin analogues and then assigned the level of use based on tertiles of the cumulative dosage in a defined daily dose, which is the assumed average maintenance dose for an adult per day.

We incorporated inpatient and outpatient diagnostic files to ascertain the history of diabetes, chronic hepatitis B, chronic hepatitis C, liver cirrhosis, hypertension, hyperlipidemia, cardiovascular disease, cerebrovascular disease, peripheral vascular disease, chronic kidney disease, and depression based on ICD-9-CM codes. We also collected other information, such as age, gender, socio-economic status, and medical resource utilization during the study period.

### 2.5. Statistical Analysis

Demographic characteristics, medication use before cancer diagnosis, comorbidities, and resource utilization for cases and matched controls in study cohorts were separately presented. We only analyzed individuals with complete data. Conditional logistic regression before and after exclusion of chronic viral hepatitis was used to estimate the crude and adjusted odds ratios (ORs) and the 95% confidence intervals (CIs) for the association between use of insulin analogues and HCC occurrence, with non-use as the reference group. Potential covariates, including socioeconomic status, comorbidities at cancer diagnosis, any use of other medications, and medical resource utilization during the follow-up period, were added into the model. All study insulin analogues were incorporated into the models, while other variables were selected by a stepwise approach, i.e., *P* < 0.10 for model entry and *P* > 0.05 for removal. Discontinuation of insulin analogues use was defined by a lack of medication refill. We divided the person-time of study drug use into current, recent (from drug discontinuation to cancer diagnosis <6 months), and past use (from drug discontinuation to cancer diagnosis ≥6 months). In the dose- and duration-response analyses, we calculated the tertile-specific ORs for the cumulative dosage (low, intermediate, and high vs. non-use) and the cumulative duration of use (short, intermediate, and long vs. non-use). Additive and multiplicative interaction analyses between study analogues and chronic viral hepatitis on HCC risks were also performed. A two-sided *P* < 0.05 was considered statistically significant. All statistical analyses were performed using SAS 9.4 (SAS Institute, Cary, NC, USA).

## 3. Results

### 3.1. Patients’ Characteristics and Univariate Analyses

Considering the first year of availability for each insulin analogue (2003 or 2004) and the presence of chronic viral hepatitis, there were four sets of study participants in our study (Figure 1, Appendix A). In the univariate analysis from 2003 to 2013 before excluding participants with chronic viral hepatitis (5832 HCC cases), patients with incident HCC were more likely to use premixed insulin analogues, beta-blockers, and diuretics; and to have comorbid liver cirrhosis, chronic hepatitis B, chronic hepatitis C, heart failure, chronic kidney disease, and a higher Charlson index (Appendix A). In the univariate analysis from 2003 to 2013 after excluding participants with chronic viral hepatitis (1237 HCC cases), HCC incidence was still positively related to any use of premixed insulin analogues (OR, 2.38; 95% CI 1.77 to 3.20) (Table 1). Any use of antidiabetic drugs, statins, and fibrates remained inversely associated with risk of HCC occurrence whether before or after excluding chronic viral hepatitis.

In the univariate analysis from 2004 to 2013 before excluding participants with chronic viral hepatitis (4573 HCC cases), HCC patients were more likely to use insulin glargine and insulin detemir, beta-blockers, and diuretics; and to have comorbid liver cirrhosis, chronic hepatitis B, chronic hepatitis C, heart failure, chronic kidney disease, and a higher Charlson index (Appendix A). In the univariate analysis from 2004 to 2013 after excluding participants with chronic viral hepatitis (952 HCC cases), the risk of incident HCC was still positively associated with use of insulin glargine and insulin detemir (OR, 1.49; 95% CI 1.06 to 2.10) (Appendix A). Any use of oral antidiabetic drugs, statins, fibrates, and angiotensin-converting enzyme inhibitors or angiotensin II receptor blockers was associated with a reduced risk of HCC occurrence.

### 3.2. Multivariate Analysis

In the multiple conditional logistic regression analysis from 2003 to 2013, incident HCC risks were independently positively associated with any use of premixed insulin analogues (adjusted OR, 1.27; 95% CI 1.04 to 1.55) before exclusion of patients with chronic viral hepatitis (Table 2). As compared to non-users, there existed positive associations of increased HCC occurrences among current users of premixed insulin analogues (adjusted OR, 1.45; 95% CI 1.12 to 1.89), those with high cumulative dosages of premixed insulin analogues (adjusted OR, 1.62; 95% CI 1.16 to 2.26), and those with long (adjusted OR, 1.80; 95% CI 1.30 to 2.51) or short duration (adjusted OR, 1.41; 95% CI 1.03 to 1.93) of premixed insulin analogues. Nonetheless, there were no more independent positive associations between the incident HCC risks and any use premixed insulin analogues (adjusted OR, 1.35; 95% CI 0.92 to 1.98) after exclusion of patients with chronic viral hepatitis. Similarly, the multivariate analysis from 2003 to 2013 showed any use of all insulin analogues to be positively associated with HCC occurrence only before exclusion of chronic viral hepatitis (adjusted OR, 1.20; 95% CI 1.03 to 1.40) (Appendix A). In the multiple conditional logistic regression analysis from 2004 to 2013 (Appendix A), HCC occurrence was not significantly associated with any use of insulin glargine or detemir, whether after excluding participants with chronic viral hepatitis (adjusted OR, 0.91; 95% CI 0.58 to 1.42) or before excluding participants with chronic viral hepatitis (adjusted OR, 1.04; 95% CI 0.84 to 1.30).

### 3.3. Interaction Analyses

There were significant multiplicative interactions between chronic viral hepatitis and premixed insulin analogues on risk of HCC after multiple adjustment (*P* = 0.010). When compared with participants who had neither chronic viral hepatitis nor any use of premixed insulin analogues, the sole presence of any use of premixed insulin analogues had a significantly higher adjusted OR of 1.51 for HCC occurrence. The addition of chronic viral hepatitis to any use of premixed insulin analogues further significantly increased the adjusted OR to 8.16 for HCC risk (Table 3). On the other hand, there was no significant multiplicative interaction between chronic viral hepatitis and insulin glargine or detemir on risk of HCC after multiple adjustment (*P* = 0.092) (Appendix A).

## 4. Discussion

This population-based study explored the relationship between insulin analogues, chronic viral hepatitis, and HCC incidence among patients with newly-diagnosed type 2 diabetes who had been prescribed at least one kind of antidiabetic agent. We have demonstrated the presence of additive and multiplicative interactions between chronic viral hepatitis and any use of premixed insulin analogues on HCC occurrence. The significantly positive association between premixed insulin analogues and HCC occurrence diminished after exclusion of patients with chronic viral hepatitis.

This large-scale nested case-control study aimed to evaluate whether chronic viral hepatitis modifies the effects of antidiabetic drugs on HCC development. Almost no previous epidemiological studies had a sample size large enough to investigate the independent association of HCC risk with insulin analogues after patients with chronic viral hepatitis had been excluded. We utilized nationwide population-based cohorts in a HCC-endemic area to determine whether chronic viral hepatitis might signify the role of some insulin analogues in liver cancer oncogenesis.

We have to address some limitations in this research. First, it is limited in generalizability to patients other than the study population. Second, this nationwide database did not provide sufficient information about body mass index, serum lipid level, smoking habits, alcohol consumption, and out-of-pocket medication history. Obesity and low triglyceride levels were reported to be positively associated with an increased risk of primary liver cancer [23,24]. Nevertheless, these factors are seldom considered to affect the clinical indication, i.e., initiation of insulin analogues for the study participants. The National Health Insurance coverage of general population in Taiwan is 99.7% as of 2015, and most of the study participants received health-insurance-covered medications for diabetes control, instead of more expensive out-of-pocket drugs [25]. Some may critique the underlying conditions of participants to be prescribed certain insulin analogues versus other anti-diabetic agents. Nonetheless, the present study did not show significant associations of any use of insulin analogues with HCC risks after exclusion of participants with chronic viral hepatitis. Actually, we intended to emphasize the intensified effect of chronic viral hepatitis on insulin analogues for HCC risks, rather than the independent role of each insulin analogue in HCC occurrence in the absence of chronic viral hepatitis. Finally, we chose a nested case-control design using a conditional logistic regression analysis rather than a survival analysis with the Cox model. However, the outcome of HCC in this large-scale nationwide nested case-control study was indeed very rare, especially in the setting without chronic viral hepatitis (Figure 1, Appendix A), which supports our original study design.

A growing number of studies have studied the association between insulin use and its HCC risk [26,27,28,29,30]. For example, a hospital-based case-control study indicated an insignificantly elevated risk of developing HCC associated with insulin use [30]. Some studies reported that insulin use was related to an increased risk of HCC development and mortality [26,27,28]. Nevertheless, these studies did not further investigate whether the increased risk persisted among each class of insulin analogues, neither did they clarify the influence of chronic viral hepatitis. Another case-control study observed increased risks for HCC in diabetic patients treated with insulin or sulfonylurea [29]. Although this study had considered hepatitis B, hepatitis C, and alcohol consumption to be confounding factors, multivariate analysis was not performed to elucidate the independent association between insulin use and the HCC risk.

Probable mechanisms linking insulin to increased HCC risks have been proposed [31,32,33]. Overexpression of insulin receptors might lead to HCC growth [31]. Hyperinsulinemia as a compensation for insulin resistance activates insulin receptor substrate-1, together with the downstream mitogen-activated protein kinase and phosphatidylinositol-3 kinase/Akt pathway, which could subsequently activate peroxisome proliferator activated receptor γ [32]. The activation of peroxisome proliferator activated receptor γ might then result in lipogenesis, adipogenesis, oxidative stress, and eventually HCC. Furthermore, hyperinsulinemia may also contribute to HCC development by inducing fibrogenesis of hepatic stellate cells, hepatocyte transformation, and angiogenesis of endothelial cells [33]. One recent study consistently demonstrated that the insulin-protein kinase and phosphatidylinositol-3 kinase/Akt-p70S6K pathways were involved in serum-enhanced cell proliferation during initial activation of hepatic stellate cells in rats [34].

Most HCC cases in Taiwan are attributable to chronic viral hepatitis [3]. Our results showed a significant interaction between chronic viral hepatitis and premixed insulin analogues on risks of HCC. Although there was neither a significant association between basal insulin analogues and HCC occurrence before exclusion of chronic viral hepatitis, nor a significant multiplicative interaction between chronic viral hepatitis and basal insulin analogues on risk of HCC after multiple adjustment, we consistently observed that chronic viral hepatitis signified the association of all insulin analogues with liver cancer risks after pooling together all kinds of insulin analogues for the multivariate analysis (Appendix A). More experimental research should explore the interplay between chronic viral hepatitis and different insulin analogues in HCC oncogenesis. It is of note that the patient’s physiological state on insulin therapy is not comparable to that of hyperinsulinemia. Actually, insulin therapy is suggested for type 2 diabetes patients who have a poor beta-cell reserve [35], which means decompensation and is not necessarily accompanied by hyperinsulinemia. In addition, the presence of hepatitis B virus may retain insulin receptors intracellularly at the endoplasmic reticulum and impede insulin receptor signaling, which could prevent compensatory liver regeneration and lead to liver disease progression [36]. Hepatitis C virus might also induce insulin resistance through insulin receptor substrate-1 serine phosphorylation and upregulated gluconeogenesis [37]. Taken together, the exclusion of chronic viral hepatitis might weaken the effect of insulin resistance and hyperinsulinemia on HCC development. Whether exogenous insulin in type 2 diabetic treatment has a similar physiological effect as endogenous hyperinsulinemia in insulin resistance warrants further investigation.

## 5. Conclusions

This nationwide population-based nested case-control study has demonstrated an intensified effect of chronic viral hepatitis on premixed insulin analogues for HCC development among newly diagnosed type 2 diabetes patients ever prescribed any antidiabetic agents. We should consider both viral- and drug-related factors in liver cancer screening programs because chronic viral hepatitis signifies the association between certain insulin analogues and HCC oncogenesis. We recommend closer liver surveillance for diabetic patients currently prescribed premixed insulin analogues with concomitant chronic viral hepatitis in general practice.

## Figures and Tables

**Figure 1 ijerph-16-02097-f001:**
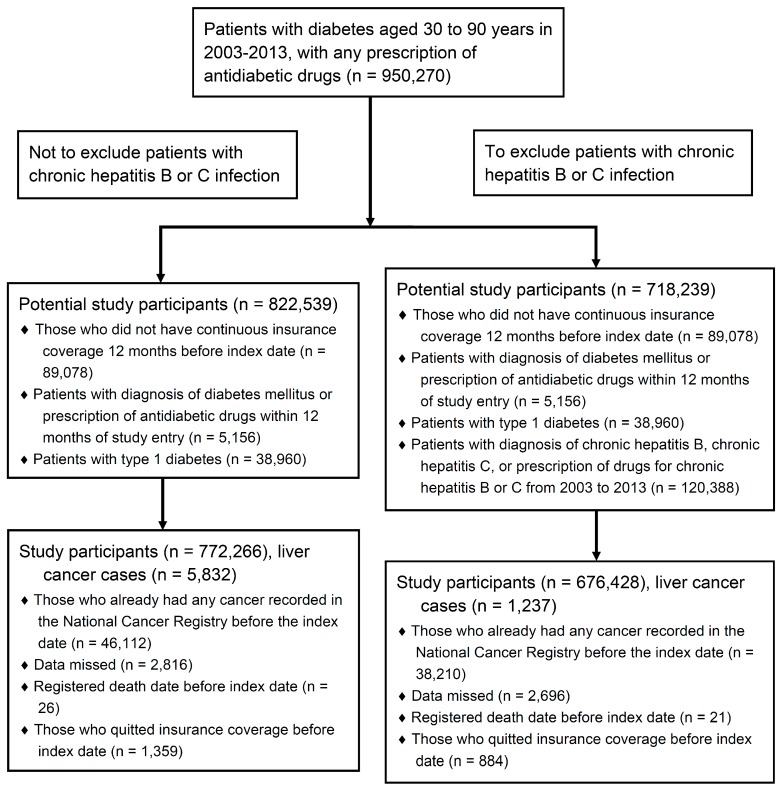
Study flowchart from 2003 to 2013 before and after exclusion of patients with chronic viral hepatitis.

**Table 1 ijerph-16-02097-t001:** Hepatocellular carcinoma cases and matched controls among newly diagnosed type 2 diabetes patients without chronic viral hepatitis prescribed any antidiabetic agents.

Factors	Cases (N = 1237)	Controls (N = 4948)	Crude OR (95% CI)
Age	64.8 (11.5)	64.5 (11.5)	−
Male	932 (75.3)	3728 (75.3)	−
Year of initiating antidiabetic agents			
2003	201 (16.3)	804 (16.3)	−
2004	179 (14.5)	716 (14.5)	−
2005	182 (14.7)	728 (14.7)	−
2006	130 (10.5)	520 (10.5)	−
2007	155 (12.5)	620 (12.5)	−
2008	115 (9.3)	460 (9.3)	−
2009	103 (8.3)	412 (8.3)	−
2010	84 (6.8)	336 (6.8)	−
2011	51 (4.1)	204 (4.1)	−
2012	32 (2.6)	128 (2.6)	−
2013	5 (0.4)	20 (0.4)	−
Socioeconomic status (monthly income in NTD)			
≤17,280	530 (42.9)	2253 (45.5)	Reference
17,281~22,800	483 (39.1)	1774 (35.9)	1.16 (1.01–1.33)
22,801~28,800	54 (4.4)	202 (4.1)	1.14 (0.83–1.57)
28,801~36,300	56 (4.5)	219 (4.4)	1.09 (0.79–1.49)
36,301~45,800	65 (5.3)	240 (4.9)	1.15 (0.85–1.55)
>45,800	49 (4.0)	260 (5.3)	0.80 (0.58–1.11)
Medication use before cancer diagnosis			
Premixed insulin analogues	73 (5.9)	127 (2.6)	2.38 (1.77–3.20)
Oral antidiabetic drugs	1058 (85.5)	4464 (90.2)	0.63 (0.52–0.76)
ACEi/ARBs	99 (8.0)	389 (7.9)	1.02 (0.81–1.29)
Beta-blockers	155 (12.5)	435 (8.8)	1.50 (1.23–1.83)
Calcium channel blockers	73 (5.9)	273 (5.5)	1.08 (0.82–1.42)
Diuretics	85 (6.9)	151 (3.1)	2.40 (1.82–3.17)
Statins	147 (11.9)	915 (18.5)	0.58 (0.48–0.70)
Fibrates	137 (11.1)	658 (13.3)	0.81 (0.66–0.98)
*Comorbidities*			
Liver cirrhosis	283 (22.9)	36 (0.7)	39.7 (27.0–58.6)
Hypertension	551 (44.5)	1934 (39.1)	1.26 (1.11–1.44)
Hyperlipidemia	140 (11.3)	837 (16.9)	0.63 (0.52–0.76)
Ischemic heart disease	134 (10.8)	446 (9.0)	1.23 (1.00–1.51)
Myocardial infarction	15 (1.2)	52 (1.1)	1.16 (0.65–2.06)
Heart failure	73 (5.9)	153 (3.1)	2.01 (1.50–2.69)
Atrial fibrillation	20 (1.6)	66 (1.3)	1.22 (0.73–2.01)
Cerebrovascular disease	117 (9.5)	488 (9.9)	0.95 (0.77–1.18)
Stroke	68 (5.5)	319 (6.5)	0.84 (0.64–1.11)
Peripheral vascular disease	9 (0.7)	32 (0.7)	1.13 (0.54–2.36)
Chronic kidney disease	60 (4.9)	103 (2.1)	2.41 (1.74–3.34)
Depression	15 (1.2)	63 (1.3)	0.95 (0.54–1.68)
Charlson index	2.6 (2.6)	1.0 (1.3)	1.56 (1.50–1.63)
Resource utilization			
Number of A1C measurements	4.8 (5.9)	4.8 (6.4)	1.00 (0.99–1.01)
Number of lipid measurements	4.3 (5.3)	4.5 (5.8)	0.99 (0.98–1.00)
Number of outpatient visits	88.2 (95.9)	78.6 (86.2)	1.00 (1.00–1.00)
Number of hospitalizations	2.0 (3.24)	1.1 (2.2)	1.15 (1.12–1.18)
Length of hospital stay >7 days	1.0 (2.0)	0.6 (1.4)	1.19 (1.14–1.24)

Data presented as mean (standard deviation) or number (percent). Insulin analogs included insulin glargine, insulin detemir, and premixed insulin analogs (insulin aspart plus insulin aspart protamine and insulin lispro plus insulin lispro protamine). ACEi, angiotensin-converting enzyme inhibitor; ARB, angiotensin II receptor blocker.

**Table 2 ijerph-16-02097-t002:** Risk of hepatocellular carcinoma associated with premixed insulin analogues among newly diagnosed type 2 diabetes patients prescribed any antidiabetic agents before and after exclusion of patients with chronic viral hepatitis.

	After exclusion of Chronic Hepatitis B or C				Before Exclusion of Chronic Hepatitis B or C			
Factors	HCCs	Controls	Crude OR	Adjusted OR ^†^	HCCs	Controls	Crude OR	Adjusted OR ^‡^
Non-use	1164	4821	1.0	1.0	5469	22,718	1.0	1.0
Any use	73	127	2.38 (1.77–3.20)	1.35 (0.92–1.98)	363	610	2.48 (2.17–2.84)	1.27 (1.04–1.55)
Current use	42	60	2.89 (1.94–4.31)	1.15 (0.67–1.98)	220	310	2.97 (2.49–3.54)	1.45 (1.12–1.89)
Recent use	9	14	2.65 (1.15–6.13)	1.41 (0.49–4.03)	36	67	2.21 (1.48–3.32)	0.70 (0.39–1.25)
Past use	22	53	1.73 (1.04–2.86)	1.63 (0.90–2.96)	107	233	1.92 (1.52–2.42)	1.25 (0.90–1.72)
Cumulative dosage								
High	26	39	2.74 (1.66–4.52)	1.62 (0.86–3.05)	139	182	3.18 (2.54–3.97)	1.62 (1.16–2.26)
Intermediate	23	48	1.99 (1.20–3.29)	1.04 (0.53–2.05)	98	229	1.77 (1.40–2.25)	1.05 (0.75–1.47)
Low	24	40	2.51 (1.50–4.19)	1.42 (0.75–2.70)	126	199	2.68 (2.13–3.36)	1.21 (0.87–1.68)
Cumulative duration								
Long	26	40	2.67 (1.62–4.39)	1.48 (0.78–2.82)	135	188	2.99 (2.39–3.75)	1.80 (1.30–-2.51)
Intermediate	25	43	2.42 (1.47–3.98)	1.28 (0.68–2.41)	92	220	1.74 (1.36–2.22)	0.72 (0.50–1.05)
Short	22	44	2.08 (1.24–3.50)	1.28 (0.65–2.52)	136	202	2.83 (2.27–3.53)	1.41 (1.03–1.93)

Premixed insulin analogues included insulin aspart plus insulin aspart protamine and insulin lispro plus insulin lispro protamine. ^†^ Adjusted for liver cirrhosis, hypertension, hyperlipidemia, cerebrovascular disease, Charlson score index, oral antidiabetic drugs, and statins. ^‡^ Adjusted for socioeconomic status, chronic hepatitis B, chronic hepatitis C, liver cirrhosis, hyperlipidemia, heart failure, cerebrovascular disease, chronic kidney disease, Charlson score index, oral antidiabetic drugs, diuretics, statins, fibrates, aspirin, number of A1C measurements, number of lipid measurements, and number of outpatient visits. HCC, hepatocellular carcinoma cases.

**Table 3 ijerph-16-02097-t003:** Interaction between premixed insulin analogues and chronic viral hepatitis on risk of hepatocellular carcinoma among newly diagnosed type 2 diabetes patients prescribed antidiabetic agents.

Premixed Insulin Analogs	Chronic Viral Hepatitis	HCCs	Controls	Crude OR	Adjusted OR ^†^
Non-use	Yes	2998	1154	22.7 (20.9–24.5)	6.99 (5.14–9.51)
Any use	No	132	534	2.16 (1.78–2.62)	1.51 (1.21–1.89)
Any use	Yes	231	76	26.5 (20.4–34.5)	8.16 (7.42–8.97)

Reference group: participants who had neither chronic viral hepatitis nor any use of premixed insulin analogues. *P* value for multiplicative interaction equal to 0.010. ^†^ Adjusted for socioeconomic status, liver cirrhosis, hyperlipidemia, heart failure, cerebrovascular disease, chronic kidney disease, Charlson score index, oral antidiabetic drugs, diuretics, statins, fibrates, aspirin, number of A1C measurements, number of lipid measurements, and number of outpatient visits. HCC, hepatocellular carcinoma cases.

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
