# Peer review of "Chronic Viral Hepatitis Signifies the Association of Premixed Insulin Analogues with Liver Cancer Risks: A Nationwide Population-Based Study"

_ijerph, 2019, doi:10.3390/ijerph16122097_

Round 1
Reviewer 1 Report
This manuscript presented some interesting data showing that both chronic hepatitis B or C and diabetes are both positively associated with HCC incidence. However, two major concerns here are required to address before publishing.
1, The authors divided the diabetes people into two groups. One group of patients took
insulin analogues and the other did not. They found that those use of premixed insulin analogues have a risk of HCC. However, they author may ignore the non-use group may have milder diabetes and thus with healthier body, which may be the main reason for low HCC occurrence rather than the use of insulin analogues.
2, As the author described, hepatitis virus or use of insulin analogues will increase the risk of HCC. However, the ORs more support that chronic hepatitis and diabetes affected HCC independently rather interaction. And since both factors are reported to be positive associated with HCC before, they author may discuss more to show the novelty of their finding.
Author Response
(Please refer to the attached Word file)

Reviewer 2 Report
The study by Chiang et al investigated whether chronic viral hepatitis would modify the associations between insulin analogues and HCC. The highlight of the study review is the finding that chronic viral hepatitis may signify the association of premixed insulin analogues in HCC. There are some points I’d like the authors to look at first:
Major points:
1. The authors branded the study to be ‘a nationwide population-based study’ in Taiwan by utilizing data sources from ‘The National Health Insurance Database in Taiwan’. However, it makes me wonder if such datasets from insurance can truly represent the entire picture of HCC in Taiwan. Above 10% (n= 89,078) of the potential study participants (without excluding chronic viral hepatitis: n = 822,539; with exclusion of chronic viral hepatitis: n = 718,239) were excluded due to discontinuation of insurance 12 months before the index date whereas the authors stated that ‘The National Health Insurance coverage of general 215 population in Taiwan is 99%’ (Line 214 - 215). Please explain the contradiction and provide references to the claim of a 99% insurance coverage rate. Will there be any significant selection bias involved? Please explain as well.
2. Line 60 -67: The authors stated that the study has related ethical approve and is exempted from patients’ written informed consents. Please include the corresponding exemption criteria from informed consents that are applicable to the present study specifically.
3. In the multivariate analyses, HCC incidence was not significantly associated with basal insulin analogues but premixed insulin analogues. I believe this deserves to be discussed.
Minor points:
1. Line 46 – 47: The cited study demonstrated an increased risk of HCC among hepatitis C virus-infected or cirrhotic patients with T2DM as compared to those without. Please specify.
2. Line 50 – 52: It’d be kind of the authors to provide a reference to the statement so as for the readers outside Taiwan to better understand the situation.
3. There are some grammatical and stylistic mistakes. Please invite a native speaker to proofread the manuscript.
Reviewer 3 Report
This paper is a compilation of data through the NHS in Taiwan. There is quite a bit of data to go through, though the conclusion could be a little stronger or reworded better. From line 267 "We should consider both viral and drug-related factors in liver cancer screening programs because chronic viral hepatitis might signify the association between certain insulin analogues and HCC oncogenesis. The use of the word "might" means that this is all inconclusive and the authors don't want to take a stand on whether the insulin analogs do or do not provide a positive correlation for HCC. Other than this wording in the conclusion, I see no other points of criticism for this work.
Round 2
Reviewer 2 Report
I'd like to thank the authors for the responses and revision. Please consider my concerns adequately addressed.